# What are the research priorities for idiopathic intracranial hypertension? A priority setting partnership between patients and healthcare professionals

Susan Mollan,[1] Krystal Hemmings,[2] Clare P Herd,[3] Amanda Denton,[2] Shelley Williamson,[4] Alexandra J Sinclair[5,6]

SM and KH contributed equally.

For numbered affiliations see end of article.

**Correspondence to**
Dr Alexandra J Sinclair;
a.b.sinclair@bham.ac.uk

## ABSTRACT

**Objective** Idiopathic intracranial hypertension (IIH) is under-researched and the aim was to determine the top 10 research priorities for this disease.

**Design** A modified nominal group technique was used to engage participants who had experience of IIH.

**Setting** This James Lind Alliance Priority Setting Partnership was commissioned by IIH UK, a charity.

**Participants** People with IIH, carers, family and friends, and healthcare professionals participated in two rounds of surveys to identify unique research questions unanswered by current evidence. The most popular 26 uncertainties were presented to stakeholders who then agreed the top 10 topics.

**Results** The top 10 research priorities for IIH included aetiology of IIH, the pathological mechanisms of headache in IIH, new treatments in IIH, the difference between acute and gradual visual loss, the best ways to monitor visual function, biomarkers of the disease, hormonal causes of IIH, drug therapies for the treatment of headache, weight loss and its role in IIH and finally, the best intervention to treat IIH and when should surgery be performed.

**Conclusions** This priority setting encouraged people with direct experience of IIH to collectively identify critical gaps in the existing evidence. The overarching research aspiration was to understand the aetiology and management of IIH.

### Strengths and limitations of this study

► This is the first collaboration of patients, carers and clinicians with experience of idiopathic intracranial hypertension (IIH) to achieve consensus on the priorities for future research.
► The James Lind Alliance (JLA) methods are patient centred and give funding bodies an unbiased agenda for research in IIH.
► Using online surveys as the main method for gathering questions for this Priority Setting Partnership (PSP) may mean that not all those with experience of IIH were aware or able to participate in the process.
► It is conceivable that possibly all the research questions gathered are not exhaustive.
► While the JLA process and IIH PSP study recommend those research priorities that are important, there is no guarantee of research funding.

of the disease documented to be rising[10] with the increasing prevalence of obesity.[7 8] In those with severely affected vision, surgery may be indicated.[1] For the majority, it can be a chronic condition, with headaches impacting on the quality of life of patients,[11] and an economic burden.[10 12]

Understanding where research should be directed was a priority for IIH UK, the leading charity for IIH in the UK. The James Lind Alliance (JLA), a UK National Institute for Health Research-supported initiative, aims to provide a transparent process that enables patients and healthcare professionals (HCP) to work together to agree on the most important uncertainties to inform the research agenda. The aim of this IIH Priority Setting Partnership (PSP) was to identify gaps in knowledge that matter most to key stakeholders (patients, carers and clinicians), and to indicate where future funding should be placed.

## INTRODUCTION

Clinical uncertainty in idiopathic intracranial hypertension (IIH) is evident, with the first consensus guidelines for investigation and management stating uncertainties in every aspect of the disease.[1] The 2015 Cochrane review concluded that there is a lack of evidence to guide pharmacological treatment.[2] There are a few published randomised clinical trials[3 4] and a small number of ongoing trials.[5 6] Research is infrequent due to the rarity of the IIH[7 8] and the lack of understanding of the underlying pathology.[9]

IIH predominantly affects overweight women of childbearing age with the incidence

**BMJ**

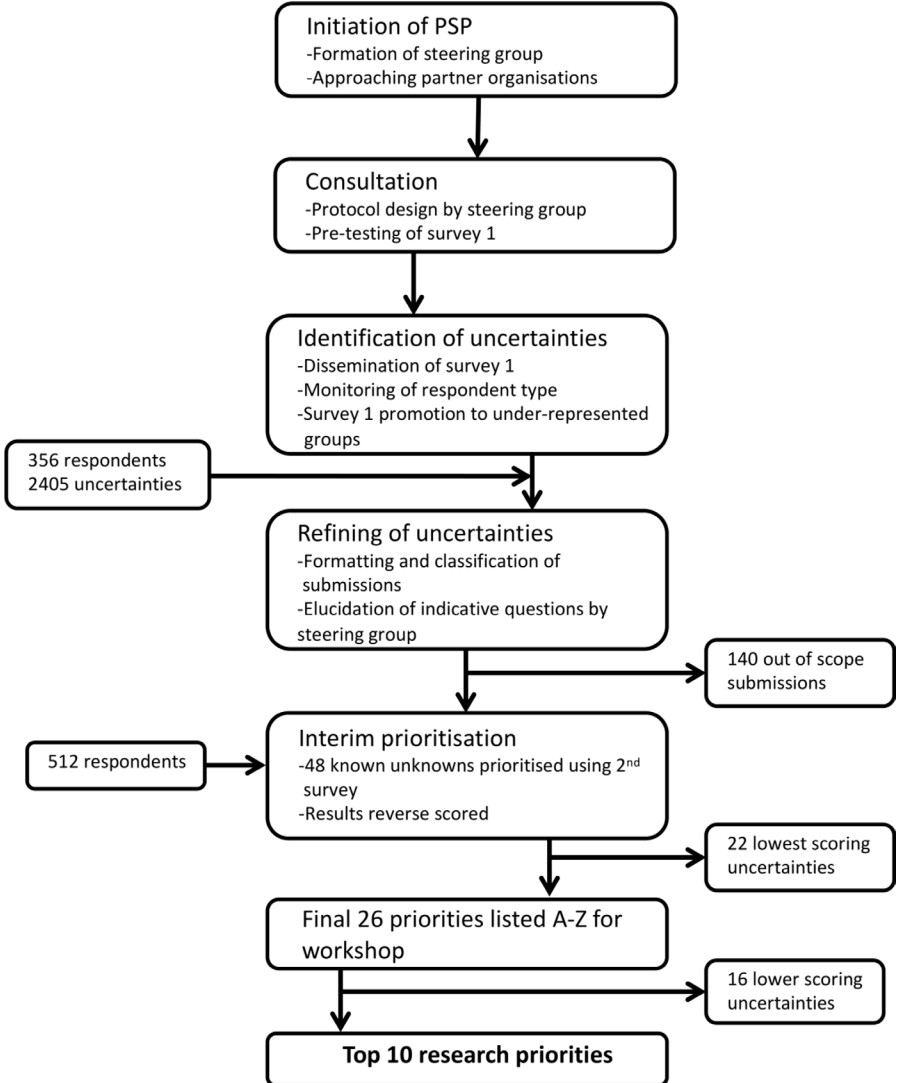

**Figure 1** Consort diagram and details of the JLA IIH PSP. IIH, idiopathic intracranial hypertension; JLA, James Lind Alliance; PSP, Priority Setting Partnership.

## METHODS
### IIH PSP process

The University of Birmingham, UK, acted as an academic partner to the IIH PSP and the process was led by the IIH UK research representative, in collaboration with the JLA (www.jla.nihr.ac.uk). A steering group with representation from IIH UK, patients and all the major specialities associated with IIH plus an independent information specialist oversaw the process (online supplementary table 1). In February 2017, key organisations accepted the invitation to become partners. They included Association of British Neurologists, British Association for the Study of Headache, British and Irish Orthoptic Society, Fight for Sight, The Royal College of Ophthalmologists, Society of British Neurological Surgeons cerebrospinal fluid (CSF) group, Shine, Neurological Alliance and The United Kingdom Neuro-Ophthalmology Special Interest Group (online supplementary table 2). The PSP stages were broadly based on the four-step process developed by the JLA (figure 1).[13]

This PSP was concerned with adult IIH only and any responses exclusively relating to children were excluded. There was limited funding for the project, and including the paediatric population would have required funding for two different work streams. It is well documented the expectantly different phenotype between adult and those prepubescent children with IIH.[14] However, responses were not limited by those who submitted and hence, those with children with IIH are likely to be included. Indeed, at the final stakeholder meeting, there was representation from carers of children with IIH. Responses concerning the classification of the disease, healthcare funding/entitlements or statements without a discernible question were excluded.

The prioritisation survey questions were constructed (online supplementary table 2) by the steering group, aided by the first guidelines in IIH where uncertainties exist around the diagnosis, investigation and management.[1] This first survey was advertised by partners (online supplementary table 3), IIH UK and steering

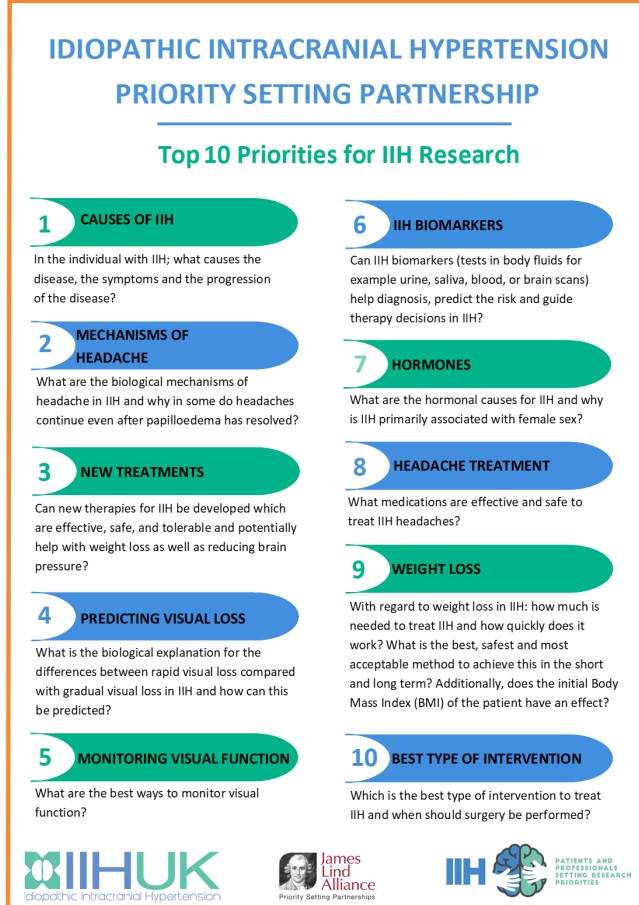

**Figure 2** Final top 10 ranked uncertainties for concerning the treatment and management of people with idiopathic intracranial hypertension (IIH).

group members. All responses were refined to understandable 'uncertainties' with the exception of those considered to be 'out of scope'. These were categorised using the UK Clinical Research Collaboration Health Research Classification System, sorted into themes and then formulated into indicative questions by steering group members, working in groups with at least one HCP and one patient representative. A literature search was conducted with the electronic databases, CENTRAL, Embase and MEDLINE, searched from inception to March 2018 for systematic reviews using strategies based on those used by Piper *et al*.[2] The 'known knowns' with reference to the appropriate literature and duplicate questions were removed. Questions were amalgamated when practical to do so. The long list was then verified by the PSP lead and discussions were held with the wider steering group if disagreements occurred.

The known unknowns were then used for the interim survey. Respondents ranked the questions, returning their top 10. The rankings were reverse scored and the total scores for the two groups, individuals with IIH, friends or carers, and HCP, were calculated separately to ensure an equal weighting. The most popular 26 questions were then taken forward, which included the top 10 for both groups, to the final workshop, with the aim of

consensus on the top 10 priorities.[13] Data relating to the PSP are available on reasonable request to IIH UK (www.iih.org.uk).

## Patient and public involvement

This research priority partnership was established by IIH UK, a charity that is run by carers and people with IIH. At each stage of the JLA process, patients and carers were equal collaborators in the design and decisions including the survey design and piloting, survey participation and the final workshop. They disseminated the surveys on the charity website and via social media. All participants were able to indicate a desire for further involvement and information about the results.

## RESULTS

The prioritisation survey generated 356 responses (figure 1). Demographic data for those with IIH is provided in online supplementary table 4 and details of HCP specialisms in online supplementary table 5. Of the 2405 generated uncertainties, 140 were out of scope. The resulting 2265 were grouped into 64 indicative questions. Sixteen were deemed to be already known or unanswerable by research, leaving 48 questions for presentation in the interim survey. Responses from 512 people were collected in a ratio of 4:1 people with IIH, friends and carers to HCP.

A final list of 26 prioritised questions was generated from the analysis of the interim survey, which included the top 10 for both groups (online supplementary table 6). The most common themes from non-HPC (healthcare professional) were why the disease develops and progresses, hormonal causes and female predominance and the conditions associated with IIH. For HCP education, the utility of biomarkers and biological mechanisms of headache were the most common. At the consensus workshop, the top 10 priorities were agreed (figure 2; online supplementary table 7).

## DISCUSSION

Understanding the most relevant research projects to fund can be challenging. It is imperative that the topics identified in a disease area have the utmost relevance to patients affected by the disease and recognised by clinicians that have a clear understanding of the clinical entity. We have undertaken a JLA PSP to establish the top 10 research areas for IIH.

The IIH JLA PSP was funded by IIH UK and set up those who have an active collaboration to improve care for people with IIH.[15] The principles and structured process outlined by the JLA was adhered to steadfastly throughout.[13] All data was maintained in a manner that could be tracked back at any point to the original questions and demographic source; this provided transparency.

A major challenge for the IIH PSP steering group was to engage all the relevant HCP (namely, neurologists,

ophthalmologists, neurosurgeons, radiologists and orthoptists). The speciality diversity brought strength to the process and allowed for a broad inclusion; however, during the final selection for the top 10, clinicians were clearly polarised by their individual specialism. There are a number of surgical treatments for fulminant visual loss in the form of CSF diversion, as directed by neurosurgeons, and optic nerve sheath fenestration, as performed by ophthalmic surgeons.[16] More recently, interventional radiologists have performed venous sinus stenting for IIH.[17] Physicians (both neurologists and ophthalmologists) use weight loss and medical therapies, such as acetazolamide and topiramate.[1 18] This mix of specialism and approach in certain patient groups, that is those at threat of visual loss or with chronic headache, led to expectantly different opinions; for example, surgeons were keen for novel interventions, whereas physicians were promoting better medical therapies.

At the interim survey, it was clear that there was a discrepancy between the non-HCP and HCP in their most popular themes, with patients keen for research into the aetiology, and HCP more commonly ranked education, biomarkers and pathological mechanisms driving headache. The top priority of the patients' group at the interim survey was the same as the final result of the consensus workshop.

Some differing opinions between non-HCP and HCP were expressed at the workshop. One issue was surrounding weight loss, seen by physicians as the only disease modifiable therapy and so a high priority for further understanding. This was a highly sensitive issue among the patients and carers present who voiced that it was not considered so important by patients. During the workshop, a collective decision was made to have a wide scope within the top 10 areas. If a topic was already featured high within the list, questions that contained a similar theme were purposely voted lower. For example, weight loss, the longer more detailed question was ranked higher than the question regarding bariatric surgery, with the reasoning that it could be answered not only by the weight loss question but also by number 10, the intervention question. For this reason, no further ranking below the top 10 should be published. Of note, two areas that did not feature in the top 10, namely, multidisciplinary clinics and an education programme. They were scored as important during the interim survey, particularly by HPC. The consensus workshop delegates agreed that although these are highly important, the PSP is intended to inform grant bodies who fund research and these areas were universally accepted to require improvement.

## Strengths

Within the feedback, people with IIH voiced that they felt their opinions were often not heard; therefore, the IIH PSP has allowed them a voice. There was a good response rate from all groups when considering how rare IIH is. Submissions with low duplication rates were not removed, a process which can introduce bias. All submitted

uncertainties were considered in the long list if they were determined to be known unknowns, including those asked by a single respondent. The data analysis followed standard protocols, though it was complicated by the use of multiple questions in the initial survey (online supplementary table 3) as each respondent could appear in up to seven separate initial categories.

## Limitations

Despite the use of identification codes, the multi-level process meant that the number of individuals contributing to the final data set could not be reasonably calculated. The project took 18 months and surveys were closed on schedule, leaving the possibility that this happened before the maximum number of respondents could contribute. Using online surveys as the main method for gathering questions for this PSP may mean that not all those with experience of IIH were aware or able to participate in the process. It is conceivable that possibly all the research questions gathered are not exhaustive. While the JLA process and IIH PSP study recommend those research priorities that are important, there is no guarantee of research funding.

## CONCLUSIONS

The IIH PSP has been an opportunity to understand the areas that are important to all. The primary topic of underlying aetiology requires work both clinically and within the basic laboratory research. Another key area highlighted by this PSP is that of mechanisms of headache in IIH. There is increasing evidence regarding the phenotype of the IIH headache, which is a challenging tradition regarding the raised intracranial pressure (ICP) headache.[19 20] Future work should explore novel therapies for headache in IIH, which is the key driver in lowering the quality of life in this patient cohort.[11] The PSP has the potential to influence the research agenda and consequently in time all area of management, from medical to surgical interventions for this currently idiopathic disease.

**Author affiliations**
[1]Department of Birmingham Neuro-Ophthalmology, University Hospitals Birmingham NHS Foundation Trust, Birmingham, UK
[2]IIH UK, Tyne and Wear, UK
[3]Institute of Applied Health Research, University of Birmingham, Birmingham, UK
[4]Idiopathic Intracranial Hypertension UK, Washington, UK
[5]Department of Metabolic Neurology, University of Birmingham, Birmingham, UK
[6]Department of Neurology, University Hospitals Birmingham, Birmingham, UK

**Correction notice** This article has been corrected since it was published online. The license type has been updated from CC BY-NC to CC BY.

**Acknowledgements** We would like to acknowledge all the people who contributed to the surveys. We would also like to acknowledge the support from the partner organisations: Association of British Neurologists, British Association for the Study of Headache, British and Irish Orthoptic Society, Fight for Sight—The Eye Research Charity, The Royal College of Ophthalmologists, Society of British Neurological Surgeons CSF group, Shine, Neurological Alliance and The United Kingdom Neuro-Ophthalmology Special Interest Group.

**Contributors** SM: interpretation of the survey results and drafting and review of the manuscript. KH: PSP patient lead, administration of both surveys and drafting and review of the manuscript. CPH: literature review, independent information specialist and drafting and review of the manuscript. AD: critical review of the manuscript. SW: organisation of the consensus final workshop and critical review of the manuscript. AJS: PSP clinical lead, interpretation of the survey results and critical review of the manuscript. All authors were steering group members and have read and approved the final manuscript.

**Funding** AJS is funded by an NIHR Clinician Scientist Fellowship (NIHR-CS-011-028) and the Medical Research Council, UK (MR/K015184/1). The JLS IIH PSP project was funded by IIH UK.

**Competing interests** None declared.

**Patient consent for publication** Not required.

**Ethics approval** The IIH UK internal review board formally reviewed the project and further ethical approval was not required. All data were anonymised and sent to the information specialist at the University of Birmingham for processing.

**Provenance and peer review** Not commissioned; externally peer reviewed.

**Data sharing statement** Any unpublished data are available from the James Lind Alliance website and from the authors.

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
