## [Reviewer comments · BMJ Open]

This paper was submitted to a another journal from BMJ but declined for publication following peer review. The authors addressed the reviewers' comments and submitted the revised paper to BMJ Open. The paper was subsequently accepted for publication at BMJ Open.

(This paper received three reviews from its previous journal but only two reviewers agreed to published their review.)

ARTICLE DETAILS

TITLE (PROVISIONAL)	What are the research priorities for idiopathic intracranial hypertension? A priority setting partnership between patients and health care professionals.
AUTHORS	Mollan, Susan; Hemmings, Krystal; Herd, Clare P; Denton, Amanda; Williamson, Shelley; Sinclair, AJ

VERSION 1 – REVIEW

REVIEWER	marek czosnyka Division of neurosurgery, university of Cambridge, uk
REVIEW RETURNED	26-Oct-2018

GENERAL COMMENTS	 1. Title should be modified: 'Research questions in IIH' 2. Limitations should be added: The list of research question is most probably not complete 3. References omit important areas of venous stenting 4. It is hardly possible to study/manage intracranial hypertension without measuring ICP and CSf dynamics. Works on invasive/non-invasive measurement of ICP should be included. Also cerebral perfusion pressure, limits of autoregulation, permanent mild brain hypoxia, etc 5. Although not very well documented, genetics is an important area 6. I do not understand why pediatric IIH was excluded
--

REVIEWER	Shanthi Viswanathan Department of Neurology Kuala Lumpur Hospital Kuala Lumpur Malaysia
REVIEW RETURNED	20-Nov-2018

GENERAL COMMENTS	Thank you for the privilege of reviewing this manuscript. This manuscript explores a very important and often not explored enough area of balancing between what health care providers as opposed to patients want to know about a particular type of disease. Thus far this sort of information from a multi-ethnic cohort (though caucasian predominant) has not been explored. More studies that engage both stake holders are needed in other disease areas too. It provides important areas of further research that needs clarification in the future for IIH. My only comment, is from all the questions formulated in the beginning with Survey one (I), maybe some clarification about
---

	survey 1 and who administered survey 1 and what criteria the committee used to reduce/whittle down the number of relevant questions to the 48 questions and finally to the 48 unknowns needs explanation. After this, the number of questions were further stratified to 26 priorities and finally to the 10 top important questions. Some clarification of the process of/criteria used by the learned survey team would help to make this manuscript even more relevant. In the conclusion part too, rather than gloss over the benefits of this information with only 3 lines, a bit more explanation about how this study will impact the future of IIH management and some clarification of how the team is going to move forwards and the foreseen benefits of the study to the nation and globally would be impactful. Aside from that I dont have any other comments as the study is meaningful, language is wonderful and the conclusions are important and very relevant to this area and will be used as a reference point in the management of IIH for many years to come.
--	---

VERSION 1 – AUTHOR RESPONSE

We would like to thank Professor Marek Czosnyka for his time, consideration of this manuscript and learned comments.

1. Title should be modified: 'Research questions in IIH'
We have amended this title to Research priorities in IIH, as the patients who have been partners in the JLA PSP and co-written this piece feel strongly that the process has prioritised the themes.

2. Limitations should be added: The list of research question is most probably not complete
Added to limitations:
It is conceivable that possibly all the research questions gathered are not exhaustive.

3. References omit important areas of venous stenting
We had not actually made in detail the various interventions for IIH which you have cited as an omission, we have therefore added paragraph on interventions in IIH to include CSF shunting, ONSF and VSS.
There are a number of surgical treatments for fulminant visual loss in the form of CSF diversion, as directed by neurosurgeons, and optic nerve sheath fenestration, as performed by ophthalmic surgeons.[14] More recently interventional radiologists have performed venous sinus stenting for IIH.[15] Physicians (both neurologists and ophthalmologists) use weight loss and medical therapies such as acetazolamide and topiramate.[1,2,3,4] This mix of specialism and approach in certain patient groups, i.e. those at threat of visual loss or those with chronic headache, led to expectantly different opinions: for example, surgeons were keen for novel interventions, whereas physicians were promoting better medical therapies

4. It is hardly possible to study/manage intracranial hypertension without measuring ICP and CSf dynamics. Works on invasive/non-invasive measurement of ICP should be included. Also cerebral perfusion pressure, limits of autoregulation, permanent mild brain hypoxia, etc

Please see supplementary table of the final 26 questions where VSS; physiology of CSF dynamics and non-invasive measures were included:
Are non-invasive intracranial pressure (ICP) measurements accurate and clinically useful?
Is cerebral venous stenosis the cause or consequence of IIH?
Is IIH caused by increased production or lack of cerebral spinal fluid (CSF) absorption?
What are the triggers for periods of high intracranial pressure (ICP) in people with IIH?

5. Although not very well documented, genetics is an important area
This is captured in the supplementary table of the final 26 questions:

Is there a genetic cause of IIH?

6. I do not understand why pediatric IIH was excluded

Added paediatric IIH was excluded, as debated with the James Lind Alliance and IIH UK. It would have required more funding to run both a paediatric and adult PSP, so as not to cause conflict as to which is the priority. We have added to the methods:

This PSP was concerned with adult IIH only and any responses relating to children were excluded. There was limited funding for the project, and including the paediatric population would have required funding for two different work streams. It is also well documented the expectantly different phenotype between adult and children with IIH.[13]

However as detailed carers and those above 16 years could reply to the surveys and at the final stakeholder meeting we did have representation

We would also like to thank Dr Shant

This manuscript explores a very important and often not explored enough area of balancing between what health care providers as opposed to patients want to know about a particular type of disease. Thus far this sort of information from a multi-ethnic cohort (though caucasian predominant) has not been explored. More studies that engage both stake holders are needed in other disease areas too. It provides important areas of further research that needs clarification in the future **hi Viswanathan for her time, consideration of the manuscript and learned comments.** for IIH.

My only comment, is from all the questions formulated in the beginning with Survey one (I), maybe some clarification about survey 1 and who administered survey 1

Added/amended: The prioritisation survey questions were constructed (supplemental table 3) by the steering group. This first survey was advertised by partners (supplemental table 2), IIH UK and steering group members.

and what criteria the committee used to reduce/whittle down the number of relevant questions to the 48 questions and finally to the 48 unknowns needs explanation.

These were categorised using the UK Clinical Research Collaboration Health Research Classification System, sorted into themes and then formulated into indicative questions by steering group members, working in groups with at least one HCP and one patient representative.

After this, the number of questions were further stratified to 26 priorities and finally to the 10 top important questions. Some clarification of the process of/criteria used by the learned survey team would help to make this manuscript even more relevant.

In the conclusion part too, rather than gloss over the benefits of this information with only 3 lines, a bit more explanation about how this study will impact the future of IIH management and some clarification of how the team is going to move forwards and the foreseen benefits of the study to the nation and globally would be impactful.

Expanded the last paragraph of the conclusions to attempt to address this.

Aside from that I dont have any other comments as the study is meaningful, language is wonderful and the conclusions are important and very relevant to this area and will be used as a reference point in the management of IIH for many years to come.

VERSION 2 – REVIEW

REVIEWER	Shanthi Viswanathan Department of Neurology, Kuala Lumpur Hospital Kuala Lumpur Malaysia
REVIEW RETURNED	06-Jan-2019
GENERAL COMMENTS	The authors have answered the questions satisfactorily. I have no further comments.